# Home-Based Exercise Training in the Recovery of Multisystem Inflammatory Syndrome in Children: A Case Series Study

**DOI:** 10.3390/children10050889

**Published:** 2023-05-16

**Authors:** Camilla Astley, Gabriela Nunes Leal, Saulo Gil, Priscila Suguita, Thais Fink, Vera Bain, Maria Fernanda Badue Pereira, Heloisa Helena Marques, Sofia Sieczkowska, Danilo Prado, Marcos Santos Lima, Camila G. Carneiro, Carlos Alberto Buchpiguel, Clovis Artur Silva, Bruno Gualano

**Affiliations:** 1Applied Physiology and Nutrition Research Group, School of Physical Education and Sport, University of Sao Paulo, Sao Paulo 01246-903, Brazil; camilla.astley@gmail.com (C.A.);; 2Rheumatology Division, Clinical Hospital, School of Medicine, University of Sao Paulo, Sao Paulo 01246-903, Brazil; 3Children and Adolescent Institute, Clinical Hospital, School of Medicine, University of Sao Paulo, Sao Paulo 05403-000, Brazil; 4Department of Radiology and Oncology, Nuclear Medicine Division, Clinical Hospital, School of Medicine, University of Sao Paulo, Sao Paulo 05403-911, Brazil; 5Food Research Center, University of Sao Paulo, Sao Paulo 05508-080, Brazil

**Keywords:** COVID-19, pediatric multisystem inflammatory disease, cardiovascular imaging, microvasculature, exercise training

## Abstract

Objective: To assess the potential therapeutic role of exercise on health-related quality of life, assessed by the Pediatric Outcomes Data Collection Instrument (PODCI), coronary flow reserve (CFR), cardiac function, cardiorespiratory fitness, and inflammatory and cardiac blood markers in multisystemic inflammatory syndrome in children (MIS-C) patients. Methods: This is a case series study of a 12-wk, home-based exercise intervention in children and adolescents after MIS-C diagnosis. From 16 MIS-C patients followed at our clinic, 6 were included (age: 7–16 years; 3 females). Three of them withdrew before the intervention and served as controls. The primary outcome was health-related quality of life, assessed PODCI. Secondary outcomes were CFR assessed by 13N-ammonia PET-CT imaging, cardiac function by echocardiography, cardiorespiratory fitness, and inflammatory and cardiac blood markers. Results: In general, patients showed poor health-related quality of life, which seemed to be improved with exercise. Additionally, exercised patients showed improvements in coronary flow reserve, cardiac function, and aerobic conditioning. Non-exercised patients exhibited a slower pattern of recovery, particularly in relation to health-related quality of life and aerobic conditioning. Conclusions: Our results suggest that exercise may play a therapeutic role in the treatment of post-discharge MIS-C patients. As our design does not allow inferring causality, randomized controlled trials are necessary to confirm these preliminary findings.

## 1. Introduction

Coronavirus disease 2019 (COVID-19) has impacted children and adolescents, with most cases being mild or asymptomatic [1]. However, multisystem inflammatory syndrome in children (MIS-C) may manifest as a rare complication 4 weeks after the SARS-CoV-2 infection, resulting in multiorgan system dysfunction, several clinical manifestations, and the need for pediatric intensive care unit (PICU) hospitalization [2,3,4,5].

The cardiovascular system is the most impacted one (80%), followed by hematologic (76%), mucocutaneous (74%), and respiratory (70%) systems [6]. The etiology of cardiovascular findings in MIS-C is likely multifactorial. The main cardiac manifestations involve arrhythmia, aneurysms, ventricular dysfunction, coronary artery dilation, conduction abnormalities, and coronary microvasculature disease (CMD) [7,8,9]. Microvascular dysfunction is considered the main driver of morbidity and mortality in COVID-19, and the severity of CMD associates with inflammatory markers, fibrin turnover, myocardial injury, and myocyte stretch [8]. 

Recently, we observed reduced coronary flow reserve (CFR) and impaired cardiopulmonary exercise capacity in a small cohort of MIS-C survivors [10]. CFR is the measure of the microvasculature’s ability to respond to a stimulus, and as such, a proxy of small vessel function [11]. Reduced CFR may predict CMD, myocardial ischemia, arrhythmia, and sudden death during strenuous activity [12]. Low aerobic fitness is a determinant of premature death and is implicated in several comorbidities, including physical dysfunction, exercise intolerance, and chronic fatigue. Given the systemic and debilitating nature of MIS-C, one could expect that this condition could also impair, to some extent, overall health-related quality of life [13]. 

To date, there is limited evidence on the management of MIS-C complications. In this scenario, exercise emerges as a safe and non-expensive tool that could be useful to rehabilitate MIS-C patients, potentially offsetting persistent impairments in cardiovascular function and physical capacity, ultimately resulting in better health-related quality of life. Physical exercise promotes an anti-inflammatory response and improves immune defense through the involvement of diverse organs (e.g., heart, lungs, skeletal muscle, brain, and intestines). The anti-inflammatory response to exercise has also been associated with a reduced risk of developing comorbidities (such as obesity) and reduced cardiovascular morbidity [14,15]. This hypothesis is supported by analogy, considering the beneficial effects of exercise in other pediatric acute or chronic conditions [16,17,18]. 

In this case series study, we aimed to explore the potential therapeutic role of exercise on several outcomes (i.e., health-related quality of life, CFR, cardiac function, cardiorespiratory fitness, and inflammatory and cardiac blood markers) in MIS-C patients. 

## 2. Material and Methods

### 2.1. Study Design and Patients

This is a case series study of a 12-wk home-based exercise intervention in children and adolescents who survived MIS-C. This study is part of a prospective cohort study aimed to explore the long-term effects of SARS-CoV-2 infection in surviving pediatric post-COVID-19 and MIS-C patients (clinicaltrials.gov NCT04659486). Data from the patient’s acute phase MIS-C were retrospectively assessed through medical records. The post-infection data were collected prospectively in an outpatient clinic for COVID-19 at the Children’s and Adolescents’ Institute of the Clinical Hospital of the University of Sao Paulo between October 2020 and January 2022. Out of the 16 MIS-C patients followed at our clinic, 4 died, 3 did not meet inclusion criteria (i.e., younger than 7 years), and 3 did not accept to participate in this exercise trial. Therefore, 6 patients were included. All patients (age: 7–16 years; 3 females) fulfilled the MIS-C diagnosis according to the Center for Disease Control (CDC) criteria [19]. Five patients had positive serologic tests (assessed by real-time reverse transcription-polymerase chain reaction (real-time RT-PCR) or antibody testing. Real-time RT-PCR to evaluate SARSCoV-2 RNA was performed on swab-collected nasopharyngeal and/or oropharyngeal samples [20]), and 1 had a negative serologic test but was exposed to a confirmed COVID-19 case within 4 weeks prior to the onset of symptoms. Demographic and clinical data at hospital admission can be seen in Table 1. Five out of six patients were admitted to PICU. Three patients required respiratory support and oxygen therapy, and three had a vasodilatory shock. The median length of stay was 11 (range: 5–18) days. The median time elapsed from discharge to the beginning of the exercise training was 5.8 (range: 1.5–10) months. Of these 6 patients who initially accepted to participate in the trial, 3 of them gave up before the intervention. We decided to follow-up on them with all the planned assessments as we found it informative to have some sort of “control data” (from non-exercised patients), despite the non-random feature of this design. 

The primary outcome was health-related quality of life, assessed by the Pediatric Outcomes Data Collection Instrument (PODCI). Secondary outcomes were CFR assessed by 13N-ammonia PET-CT imaging, cardiac function by standard echocardiography, cardiorespiratory fitness, and inflammatory and cardiac blood markers (C-reactive protein, D-dimer, fibrinogen, troponin T, hemoglobin, lymphocyte and platelet count, urea, creatinine, alanine, and aspartate transaminase). 

### 2.2. Ethics

The protocol was approved by the National and Institutional Ethical Committee of Clinical Hospital, CAAE: 37460620.8.0000.0068. Patients and guardians signed an informed consent before participants’ enrollment, and the study was conducted according to the Declaration of Helsinki. 

### 2.3. Exercise Training Program

The 12-week, 3-times-a-week, home-based exercise program used herein was thoroughly described elsewhere [21,22]. In brief, the bouts had two components. The first one included aerobic exercise predominantly, such as jumping jacks, skipping, flexibility, and mobility exercises. The second component included bodyweight exercises (push-ups, air squats, lunges, crunches, and planks). Exercise sessions occurred 3 times a week; one weekly session was supervised online by a fitness trainer, whereas the other two were unsupervised, with patients being advised to provide feedback to the staff upon the completion of the training bout. The training monitoring was conducted through WhatsApp^®^, Zoom^®,^ or Google Meets^®^ apps. The exercise intensity was assessed through the Children’s OMNI scale of perceived subject exertion (PSE) [23]. Progression was carried out every 4 weeks using the OMNI scale (week 1–4, PSE: 5–6; week 4–8, PSE: 6–7 and week 8–12, PSE: 7–8), modifying the number of repetitions (10 to 15), sets (3 to 4), and/or duration of the sets (30 to 45 s). Patients received instructional videos, photos, and “gifs” describing the exercise program. A video call was conducted prior to program initiation to provide details on the program and collect information on the patient’s health status. We assessed adherence to the intervention using a training log. 

### 2.4. Pediatric Outcomes Data Collection Instrument (PODCI)

To assess health-related quality of life, we used PODCI, which evaluates functional health status through an 83-86 item questionnaire [24]. This questionnaire consists of scores: four encompassing physical function (upper extremity and physical functioning, transfer and basic mobility, sports, and physical functioning) and two assessing psychological well-being (pain/comfort and happiness), and a PODCI global function score (0–100). Lower scores are indicative of a lower health-related quality of life. A radar plot was created using absolute values from pre and post-periods for each patient and each PODCI domain (upper extremity and physical functioning, transfer and basic mobility, sports and physical function, pain/comfort, and happiness). All radar plots were generated using the *fmsb* package (v. 0.7.5; Nakazawa, M., 2023) in the environment R (version 3.5.3; R Core Team 2020).

### 2.5. Coronary Flow Reserve Imaging Protocol by 13N-Ammonia PET-CT (13N PET-CT)

Cardiac positron emission tomography-computed tomography (PET-CT) is the gold standard noninvasive test for myocardial blood flow (MBF). The ratio of the MBF measured at maximal vasodilation over the MBF at rest is referred to as myocardial flow reserve (MFR) or coronary flow reserve (CFR), which is a measure of the vasodilatory reserve of the myocardium. Reduced CFR, in the absence of flow-limiting coronary artery disease, is believed to reflect dysfunction in the myocardial microvasculature and result in ischemia [11,25]. Protocol details can be seen in our previous work [10].

CFR was calculated as the ratio of stress MBF over the rest MBF (for each left ventricle (LV) segment (see Table 2), CFR at right coronary artery (RCA), left circumflex artery (LCX), left anterior descending (LAD), and the 17-segment model according to the American Society of Nuclear Cardiology recommendations [26] were considered abnormal when <2, borderline between 2 and 2.5, and normal when >2.5 [27].

Standard transthoracic echocardiography was performed according to the recommendations of the American Society of Echocardiography and included M-mode, two-dimensional imaging, conventional, and tissue Doppler evaluation at the septal and lateral mitral annulus [28]. The equipment used was a Philips Affiniti 70 (Andover, MA 01810 USA), with multifrequency transducers (S 5–1 and S 8–3 MHz). Cardiac chamber dimensions were obtained using two-dimensional mode, and left ventricle ejection fraction (LVEF) was calculated by Simpson’s method (normal LVEF ≥ 55%) [28].

The z-score values for the measures were calculated according to the “Boston Children’s Hospital z-score system” [29]. LV mass (g) was estimated using Devereaux’s formula according to the Penn convention and indexed for height (m) raised to an exponential power of 2.7; we used Omni calculator [28]. LV hypertrophy was diagnosed whenever the LV mass index was greater than the 95th percentile for sex and age, according to Khoury et al. [30]. The evaluation of LV diastolic function included mitral E/e’ ratio, with e’ being the average of values obtained by tissue Doppler at the septal and lateral annulus (normal E/e’ < 14) [28]. Right ventricular (RV) systolic function was assessed by tricuspid annular plane systolic excursion (TAPSE). RV systolic dysfunction was detected when the TAPSE z-score was less than −2 [31]. E and A wave velocities, E⁄A wave ratio, DT, and E⁄E’ ratio (diastolic filling pressure surrogate) will also be measured. Diastolic dysfunction was classified as mild, moderate, or severe according to current guidelines [32]. Diastolic function will be considered impaired if there is evidence of left atrial enlargement or abnormal filling pressure. 

### 2.6. 2 DST Echocardiography and Speckle Tracking

The main principle of 2DST is that each segment of myocardial tissue displays a specific pattern of gray values in the ultrasound image, commonly referred to as a speckle pattern. Tracking this acoustic pattern during the cardiac cycle enables the observer to follow the myocardial motion and to directly assess ventricular deformation [33]. To evaluate segmental LV longitudinal systolic strain, two-dimensional harmonic image cine-loop recordings of apical four-, three-, and two-chamber views were acquired and stored digitally for analysis. A sector scan angle of 30–60° and frame rates of 60–90 Hz were chosen. A good-quality electrocardiogram signal was obtained simultaneously. The endocardial and epicardial tracing was automatically generated by the computer algorithm and manually adjusted to cover the whole myocardium wall, when necessary (QLabTM software, Philips) [33]. The extent of myocardial strain in the longitudinal direction throughout the cardiac cycle was computed as percentages (absolute values), represented by the global longitudinal strain (GLS%).

### 2.7. Cardiopulmonary Exercise Test

A symptom-limited maximal cardiopulmonary exercise test was carried out on treadmill. CPETs were performed on a treadmill with an intensity-graded, maximal effort protocol and continuous gas exchange (Metalyzer IIIb/breath-by-breath). All tests in this study were performed by the same intrahospital laboratory at controlled room temperature with individuals in an upright position (20–23 °C). Test termination was determined by volitional exhaustion, and maximal effort was confirmed by a peak respiratory exchange ratio > 1.10, maximal heart rate > 95% age/gender-predicted values, or maximum rating of perceived exertion (RPE) [34]. The outcomes were absolute, and % predicted values of oxygen consumption at ventilatory anaerobic threshold (VO_2VAT_) (mL/kg/min) and VO_2peak_ (mL/kg/min). Twelve-lead ECG and gas exchange measurements were recorded continuously. Peak VO_2_ was determined as the mean value of VO_2_ during the final 30 s of the graded exercise test. The following variables were obtained breath-by-breath and expressed as 30 s averages: pulmonary oxygen uptake (VO_2_ mL·kg^−1^·min^−1^ standard temperature and pressure, dry); pulmonary ventilation (V_E_; L/min body temperature and pressure, saturated); end-tidal carbon dioxide pressure (PetCO_2_; mmHg), ventilatory equivalent ratio for carbon dioxide (V_E_/VCO_2_), and ventilatory equivalent ratio for oxygen (V_E_/VO_2_).

Other CPET variables such as the peak oxygen consumption (VO_2peak_), oxygen consumption at the ventilatory anaerobic threshold (VO_2VAT_), V_E_/VCO_2_ slope, the lowest V_E_/VCO_2_ and respiratory exchange ratio (RER), chronotropic reserve (CR) and heart rate peak (HRP) were analyzed as previously described [34]. For all dependent variables, reference values from healthy children sorted by age and sex groups, whenever available, were used for identifying a normal exercise capacity [35,36]. 

### 2.8. Statistical Analysis

Categorical data were reported as percentages and continuous data as mean ± standard deviation (SD). Fisher’s exact test was used to compare the frequency of coronary flow reserve (abnormal < 2, borderline between 2 and 2.5, and normal when >2.5), as seen in Table 2. The significance level was set at *p* ≤ 0.05.

## 3. Results

### 3.1. Exercised Patients

Figure 1 displays the flowchart of patients followed in our tertiary hospital. 

Patient I is a 16-year-old female who presented with fever, conjunctivitis, arterial hypotension, abdominal pain, and diarrhea, with no need for ICU admission (Table 1). The patient attended 83% of the supervised exercise sessions and 62% of the online sessions. After the intervention period, the global PODCI score improved by 18% (Figure 2), with the greatest improvements being seen in the pain score (194%). After exercise, abnormal CFR reduced from 12/20 to 2/20 segments, *p* = 0.0222 (Table 2), whereas normal CFR significantly increased from 23/20 to 12/20, *p* = 0.0079. LVEF% improved from 56% to 64% (Figure 3).

Noticeably, the time to exhaustion and VO_2peak_ increased by 22% and 14%. In addition, the chronotropic reserve increased by 21% and HRP by 10% (Table 3). D-dimers reduced by 16%, whereas the other laboratory markers were within the normal range throughout the intervention.

Patient II is a 7-year-old male who presented with fever, conjunctivitis, arterial hypotension, shock, and abdominal pain, requiring PICU admission, respiratory support, and oxygen therapy (Table 1). The patient completed 100% of both supervised and online exercise sessions. After the intervention period, the global PODCI score improved by 32% (Figure 1), with greater improvements seen in the upper extremity and happiness scores (203% and 157%). 

Despite the severity of the case, following hospital discharge, CFR values were all *p* > 0.05 (Table 2), and cardiac function were normal on the interim visits with no change in medication (acetylsalicylic acid 100 mg/day). The time to exhaustion and VO_2peak_ increased by 50% and 13%. In addition, the chronotropic reserve increased by 34% and the HRP by 4% (Table 3). Inflammatory markers and other laboratory findings remained within the normal range throughout the follow-up period; however, d-dimers values dramatically reduced (9978%), stabilizing at the normal range.

Patient III is a 9-year-old female who presented with fever, arterial hypotension, shock, and abdominal pain, with a need for ICU admission (Table 1). The patient compliance to the intervention was 100% to the supervised sessions, but only 37% to the online exercise sessions. After the intervention period, the global PODCI score improved by 37% (Figure 1), with greater values seen for upper extremity and sports score delta (59% and 65%). 

Before exercise, 13N PET-CT exam showed homogeneous rest but heterogeneous stress perfusion with perfusion defects developed in the slightly dilated left ventricular cavity, suggesting stress-induced myocardial ischemia associated with CFR < 2.0 in 57% and borderline 43%, with no normal values. PIII, unfortunately, did not attend the post-intervention 13N PET-CT exam due to personal reasons. 

At baseline, 2DST echocardiography showed a slight alteration in TAPSE (−2.07) and reduced LVEF (54%), and the speckle tracking exam showed reduced GLS (−17.4%). Despite after exercise, the standard echocardiography showed normal values for cardiac function, including GLS% (Figure 3), the speckle tracking exam showed an impairment of the strain in the basal and middle segments of the infero-septal and inferior walls of the left ventricle (−13%). PIII had impaired time to exhaustion and VO_2peak_ (−7% and −5%). In addition, the chronotropic reserve reduced −15% and HRP −6% (Table 3). Nonetheless, inflammatory and cardiac markers improved (i.e., CRP, d-dimers, fibrinogen, and troponin T), whereas the other laboratory markers remained unaltered.

### 3.2. Non-Exercised Patients

Patient IV is an 8-year-old female who presented with fever, abdominal pain, diarrhea, and no echocardiography abnormalities at the acute phase (Table 1). At the final evaluation, the Global PODCI score was reduced −15% (Figure 1), with the greatest reduction being in pain score (−64%). At baseline, all CFR values were normal. At the final evaluation, 6/20 CFR values were abnormal (*p* = 0.0202), 6/20 borderline (*p* = 0.0202), and 8/20 within normal values (*p* = 0.0001) (Table 2). Additionally, cardiorespiratory fitness (e.g., time to exhaustion and VO_2peak_) reduced by 18% and 19% throughout the follow-up period. Cardiac function and laboratory markers remained within normal values (Table 3).

Patient V is an 11-year-old male with type I diabetes who presented with fever, conjunctivitis, hypotension, shock, and abdominal pain. He needed ICU admission and respiratory support, and echocardiography abnormalities were detected at the acute phase (Table 1). At the final evaluation, the Global PODCI score improved by 5% (Figure 1), whereas the others score did not change. Abnormal CFR values did not change, and borderline values raised from 1/20 to 6/20 (*p* = 0.0915), whereas normal values reduced from 18/20 to 13/20 (*p* = 0.1274). Time to exhaustion and VO_2peak_ values improved by 8% and 15%, respectively. Cardiac function and laboratory markers remained in the normal range.

Patient VI is an 11-year-old male who presented with fever, conjunctivitis, arterial hypotension, shock, abdominal pain, and diarrhea. He needed ICU admission and respiratory and oxygen support, but no echocardiography abnormalities were seen in the acute phase (Table 1). At baseline, the 2DST echocardiography and speckle tracking showed normal cardiac function; however, at the final evaluation, the 2DST echocardiography showed normal values for cardiac function, including GLS (19.9%), whereas the speckle tracking exam showed impairment in the strain at the basal segment of the anteroseptal wall was −12% and the anterior wall −15%. The strain in the mid-segment of the inferoseptal and inferolateral walls were −15% and −16%. In line with these findings, abnormal CFR s increased from 0 to 8/20 (*p* = 0.0033), borderline significantly increased from 9/20 to 12/20 (*p* = 0.0012), and normal values were reduced from 11/20 to 0/20 (*p* = 0.0001). Cardiorespiratory fitness and laboratory markers remained unchanged. 

## 4. Discussion

To our knowledge, this was the first study to demonstrate the effects of an exercise training program on health-related quality of life, microvascular and cardiac function, cardiorespiratory fitness, and inflammatory and cardiac blood markers in MIS-C survivors. To date, our findings suggest that exercise may positively impact the health-related quality of life, coronary flow reserve, cardiac function, and aerobic conditioning. 

Pediatric-COVID patients have shown reduced health-related quality of life and at least one symptom related to long-COVID 4 months after hospital discharge [37]. Moreover, biopsychosocial stressors imposed by the pandemic may increase the signs and symptoms such as anxiety, depression, weight gain, physical inactivity, and possibly increase cardiovascular risk [38,39]. In this scenario, exercise training emerges as a potential therapeutic tool to mitigate the impaired health-related quality of life and other possible biopsychosocial side effects of the COVID-19 pandemic period in children’s and adolescents’ health. 

Overall, patients assessed herein, particularly those included in the exercised group, showed poor health-related quality of life. However, exercise training programs lead to an improvement in these parameters for all patients, indicating a key role of exercise in the management of MIS-C patients. Moreover, patients who followed the standard of care did not have low quality-of-life scores in the initial assessment, preventing significant improvements in the domains assessed through the PODCI.

Cardiovascular complications arising from acute COVID-19 have been shown in the literature [40], inclusive in pediatric patients [38,41], which may increase the risk of cardiac events. In this scenario, the safety and efficacy of exercise training programs should be considered. In the current study, no mild, moderate, or severe side effects related to home-based exercise training program was reported during supervised or online sessions. Furthermore, it is noteworthy that exercised patients demonstrated an improvement in the microvascular and cardiac function (CFR, LVEF%), except for patient III, that did not perform the post-test PET-CT exam and showed defects in the segmental cardiac strain at post-test. In contrast, all non-exercised patients showed a higher frequency of abnormal CFR segments, indicating the absence of improvements in microvascular and cardiac function. Our hypothesis is that the adherence of PIII (37.5%) to supervised sessions was low, and the possible cardiovascular effects and adaptations related to physical exercise were not achieved. These findings are quite relevant since the home-based exercise training program showed to be safe and efficient in improving cardiovascular and cardiac function, lowering the possible cardiovascular risk in post-discharged MIS-C patients.

Some studies have reported impaired microvasculature in COVID-19 and MIS-C patients [42]. The mechanisms underlining the pathophysiology of disease are not fully elucidated. Myocardial ischemia and, hence, angina result from excessive myocardial oxygen that overcomes the oxygen supply. In some cases, transient ischemia can yield to persistent dysfunction following the restoration of flow. It is also possible that persistent asymptomatic ischemia leads to LV dysfunction, which mimics nonischemic causes of heart failure [43].

Blomster et al. analyzed the correlation between CFR, exercise capacity, and cardiac systolic and diastolic function in 400 adult patients with coronary artery disease (CAD). The authors showed that maximal exercise capacity is dependent on coronary CFR in non-obstructive CAD, emphasizing the significance of microvascular circulation on cardiac performance [44].

The abnormal CFR detected by PET-CT in our surviving MIS-C patients (PIII and PVI), mirrored by segmental peak systolic longitudinal strain reduction, seems to indicate residual damage in the coronary microcirculation as a consequence of infection. Importantly, microcirculatory impairment (segmental strain reduction) was detected at the standard echocardiography even in the absence of coronary artery aneurysm. 

13N PET-CT-measured impaired vasodilation capacity is associated with the increased risk of progression of congestive heart failure and mortality in adults with idiopathic cardiomyopathy [45]. Therefore, the long-term impacts of myocardial microvasculature compromise in MIS-C patients should be investigated. 

The strengths of this study involved the investigation of patients with a rare condition and the use of gold standard methods to assess cardiac-and fitness outcomes, as well as the implementation of a newly developed home-based exercise training program aimed at treating MIS-C patients. The main limitations included the low number of patients enrolled (owing to the rarity of the condition) and the lack of a control group without MIS-C, which hampers establishing causation and limits possible insights on the natural course of the syndrome. 

To conclude, our results may suggest that exercise could be a useful tool in the management of MIS-C patients. Randomized controlled studies are needed to confirm or refute these exploratory findings. 

## Figures and Tables

**Figure 1 children-10-00889-f001:**
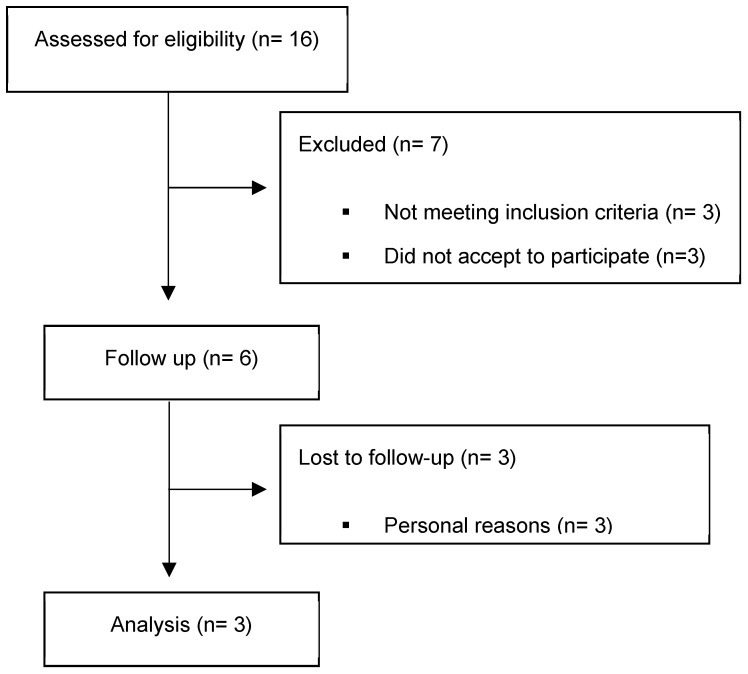
Flowchart of MIS-C patients from our tertiary hospital.

**Figure 2 children-10-00889-f002:**
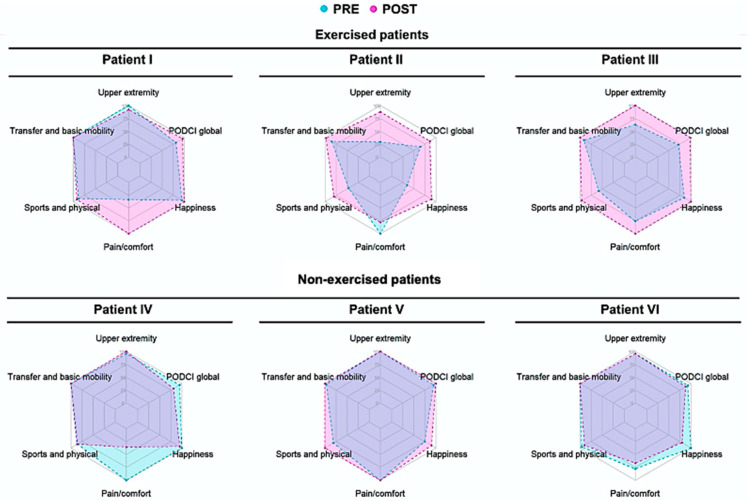
Radar-plot: Healthy-related quality of life domains in exercised and non-exercised MIS-C patients.

**Figure 3 children-10-00889-f003:**
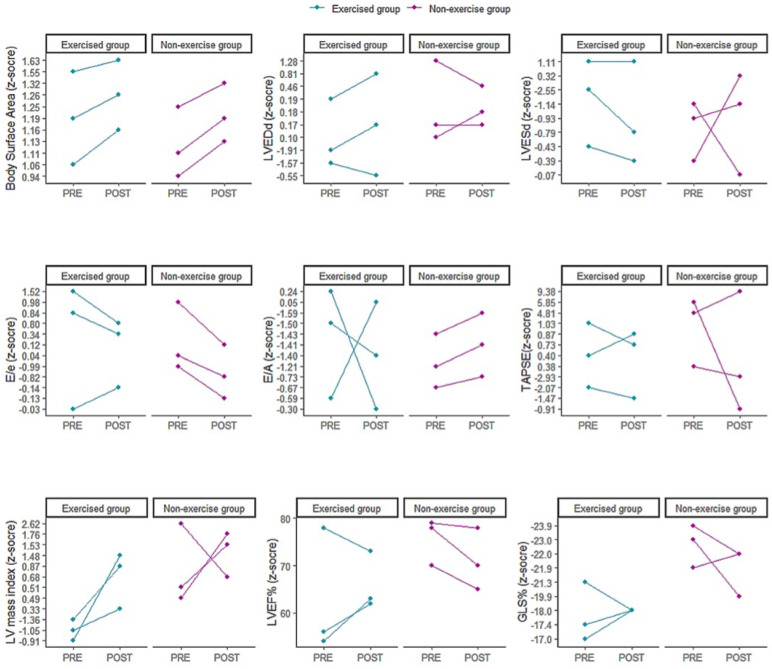
Echocardiographic parameters responses after exercise training in MIS-C patients. Abbreviations: LVEDd, left ventricle end-diastolic dimension; LVESd, left ventricle end-systolic dimension, E/A, early to late diastolic flow velocity ratio; TAPSE, tricuspid annular plane systolic excursion; LVEF, left ventricular ejection fraction; GLS, global longitudinal strain.

**Table 1 children-10-00889-t001:** Clinical features among MIS-C patients during the acute phase.

	Exercised Patients			Non-Exercised Patients		
Patient’s Characteristics	PI	PII	PIII	PIV	PV	PVI
Sex	Female	Male	Female	Female	Male	Male
Age (years)	16	7	9	8	11	11
Previously medical history	No	No	No	No	Yes (T1D)	No
Height (cm)	156	126	145	135	139	138
Weight (kg)	60.3	33.7	38.6	38.4	31.5	29.3
BMI (kg·m^−2^)	24.7	21.2	18.3	21.1	16.3	15.3
Adherence to 12-wk home-based exercise program (supervised/online)	83.3/62.5	100/100	100/37.5	*	*	*
Signs and symptoms at admission						
Fever (days)	Yes (12)	Yes (8)	Yes (12)	Yes (1)	Yes (3)	Yes (5)
Conjunctivitis	Yes	Yes	No	No	Yes	Yes
Arterial hypotension	Yes	Yes	Yes	No	Yes	Yes
Shock	No	Yes	Yes	No	Yes	Yes
Abdominal pain	Yes	Yes	Yes	Yes	Yes	Yes
Diarrhea	Yes	No	No	Yes	No	Yes
Echocardiogram abnormalities	Yes *	Yes ^#^	Yes *	No	Yes ^$^	No
Treatment						
ICU admission	No	Yes	Yes	Yes	Yes	Yes
Length of stay at hospital (days)	3	14	18	5	10	12
Respiratory support/Oxygen therapy	No/No	Yes/Yes	No/No	No/No	Yes/No	Yes/Yes
Anti-inflammatory treatment	No	Yes (mPRED)	Yes (mPRED)	No	Yes (mPRED)	No
Immunoglobulin treatment	First dose 2 g/kg	First and second dose 2 g/kg	First dose 2 g/kg	First dose 2 g/kg	First dose 2 g/kg; second dose 1 g/kg	First dose 2 g/kg

Abbreviation: T1D: type 1 diabetes; BMI: body mass index; ICU: intensive care unit; mPRED: Methylprednisolone. * Pericardial effusion; ^#^ Echogenicity of coronary arteries, without dilation; ^$^ Mild ectasia of right and left coronary arteries.

**Table 2 children-10-00889-t002:** Distribution of LV segments according to 13N PET-CT CFR categories for each MIS-C patient pre and post-12-wk exercise training program.

Exercised Patients				
		Pre	Post	Fisher’s Test
I	CFR abnormal	12/20	2/20	0.0022
	CFR borderline	5/20	6/20	1.0000
	CFR normal	3/20	12/20	0.0079
II	CFR abnormal	0/20	0/20	1.0000
	CFR borderline	1/20	6/20	0.0915
	CFR normal	19/20	14/20	0.0915
III *	CFR abnormal	11/20	-	-
	CFR borderline	9/20	-	-
	CFR normal	0/20	-	-
Non-exercised patients				
IV	CFR abnormal	0/20	6/20	0.0202
	CFR borderline	0/20	6/20	0.0202
	CFR normal	20/20	8/20	0.0001
V	CFR abnormal	1/20	1/20	1.0000
	CFR borderline	1/20	6/20	0.0915
	CFR normal	18/20	13/20	0.1274
VI	CFR abnormal	0/20	8/20	0.0033
	CFR borderline	9/20	12/20	0.0012
	CFR normal	11/20	0/20	0.0001

Abbreviations: LV: left ventricle; CFR: coronary flow reserve. * PIII did not complete the post-test PET-CT exam due to personal reasons.

**Table 3 children-10-00889-t003:** Cardiopulmonary exercise test and laboratory findings among MIS-C patients pre and post-12-wk home-based program.

	Exercised Patients			Non-Exercised Patients		
	PI	PII	PIII	PIV	PV	PVI
Cardiopulmonary exercise test						
Time to exhaustion (min)						
Pre	9.0	7	9.2	11.3	11.4	10
Post	11.0	10.5	8.5	9.3	12.3	10
Δ, %	22.2	50.0	−7.61	−17.7	7.89	0
VO_2peak_ (mL·kg^−1^·min^−1^)						
Pre	22.5	18.3	30.8	35.2	42.5	37.1
Post	25.6	20.7	29.3	28.4	48.9	38.7
Δ, %	13.8	13.4	−4.87	−19.2	15	4.39
VO_2VAT_ (mL·kg^−1^·min^−1^)						
Pre	9.66	12.9	11.1	14.6	16.2	19.2
Post	9.73	15.6	15.8	12.9	24.8	16.7
Δ, %	0.72	20.9	−29.6	−11.8	52.8	−13.1
% Predicted VO_2peak_ (<80% abnormal)						
Pre	48.7	37.1	70.1	64.5	78.7	69.4
Post	55.5	42.1	66.7	62.8	90.5	72.5
Δ, %	13.8	13.4	−4.87	−2.54	14.9	4.39
V_E_/VCO_2_ slope (units) (>31 abnormal)						
Pre	38.8	31.6	33.6	33.0	39.5	31.3
Post	42.2	32.8	36.9	36.1	39.6	37.8
Δ, %	8.76	3.80	9.82	9.39	0.25	20.7
PetCO_2_ rest (mmHg) (<35 abnormal)						
Pre	24	32	31	35	37	35
Post	27	29	30	35	36	30
Δ, %	12.5	−9.38	−5.88	0	−2.7	−14.3
O_2_ pulse peak (mL/beat) (<14 abnormal)						
Pre	8	4	7	10	7	7
Post	10	5	7	8	9	7
Δ, %	25	25	0	−20	28.5	0
Resting heart rate (beats/min)						
Pre	91	107	120	93	100	115
Post	94	99	114	94	91	90
Δ, %	9.5	−7.4	−5	1.1	−9.0	−21.7
Heart rate peak (beats/min)						
Pre	178	134	195	188	195	170
Post	195	139	183	208	195	171
Δ, %	9.5	3.7	−6.1	10.6	0	0.6
Chronotropic reserve (%) (abnormal < 80%)						
Pre	87.9	35.5	112	94.6	106.7	75.3
Post	106.3	47.6	94.5	118.8	106.1	82.7
Δ, %	20.9	34.0	−15.5	25.5	−0.58	9.70
Laboratory data (normal range)						
C-reactive protein (0.3–10 mg/L)						
Pre	0.30	0.42	4.85	0.30	0.30	0.57
Post	0.71	1.06	2.90	0.32	0.60	0.30
Δ, %	136.6	152.3	−40.2	6.67	100	−47.3
D-dimers (≤500 ng/mL)						
Pre	794	97572	1691	*	271	349
Post	665	215	640	*	326	337
Δ, %	−16.2	−9978	−62.1	*	20.3	−3.44
Fibrinogen (200–400 mg/dL)						
Pre	311	190	465	*	304	263
Post	349	315	340	239	257	252
Δ, %	12.2	65.8	−26.8	*	−15.4	−4.18
Troponin T (<0.004 ng/mL)						
Pre	0.005	0.004	0.004	*	0.003	0.007
Post	0.003	0.004	0.003	*	0.003	0.003
Δ, %	−40.0	0	−25.0	*	0	−57.1
Hemoglobin (11.5–15.5 g/dL)						
Pre	10.9	14.6	12.7	12.7	12.3	11.2
Post	10.8	13.7	13.4	12.8	12.2	12.3
Δ, %	−0.92	−6.16	5.51	0.79	−0.81	9.82
Lymphocyte count (1.5^−7^ × 10^9^/L)						
Pre	3.29	6.29	3.51	2.90	1.72	2.10
Post	2.40	5.18	2.53	2.56	1.55	2.28
Δ, %	−27.0	−17.6	−27.9	−11.7	−9.88	9.09
Platelet count (150^−400^ × 10^9^/L)						
Pre	413	473	409	336	194	376
Post	368	359	405	314	178	449
Δ, %	−10.9	−24.1	−0.98	−6.55	−8.25	19.4
Urea (7–20 mg/dL)						
Pre	23	27	18	18	23	41
Post	27	24	12	17	23	24
Δ, %	17.3	−11.1	−33.3	−5.56	0	−41.6
Creatinine (0.59–1.53 mg/dL)						
Pre	0.61	0.47	0.46	0.46	0.48	0.44
Post	0.77	0.54	0.54	0.50	0.53	0.52
Δ, %	26.2	14.9	17.3	8.70	10.4	18.1
Alanine transaminase (7–55 U/L)						
Pre	20	19	36	20	19	25
Post	15	19	26	15	13	23
Δ, %	−25.0	0	−27.7	−25.0	−31.6	−8.0
Aspartate transaminase (8–33 U/L)						
Pre	13	11	26	12	9	12
Post	9	12	19	11	7	10
Δ, %	−30.7	9.09	−26.9	−8.33	−22.2	−16.6

Abbreviations: VO_2peak_: peak oxygen consumption; VO_2VAT_: oxygen consumption at ventilatory anaerobic threshold; V_E_/VCO_2_: PetCO2; Δ = Delta.

## Data Availability

Access to de-identified data or related documents can be requested through submission of a proposal with a valuable research question, necessary data protection plan, and ethical approvals. Data requests should be addressed to the corresponding author.

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
