# Peer review of "Home-Based Exercise Training in the Recovery of Multisystem Inflammatory Syndrome in Children: A Case Series Study"

_children, 2023, doi:10.3390/children10050889_

Round 1

Reviewer 1 Report

I am grateful to have had the opportunity to read carefully this interesting object of study. It is very stimulating to propose the practice of exercise to any type of patient, no matter how affected they may be. It is interesting as communications of several cases, but I think the conclusions should either be removed or changed, because there are many methodological problems to reach any conclusion, and put hypothetical conclusions make the work loses strength. I would communicate only the data for the readers' assessment.

First of all, I would remove "Exercise Training" from the title, and leave only: "home-based physical exercise" because the essential characteristics of a training program such as individualization and intensity of the load are not communicated. There is hardly any mention of frequency and time of exercise (with many limitations), nor is the type and mode of exercise specified with sufficient rigor.

The reasons for leaving only the communication of data without interpreting them in the form of conclusions is because there were very few patients, it is not known why some did not want to follow the program (were they more affected?), it is not clear but it seems that there are different lags for each patient from the time they become ill until the tests are performed. It is not explained whether or not the patients who died were doing home exercise. The clinical significance of the observed changes is not explained, nor is the interference of medications. Nor is there any statistical treatment, despite the fact that p (?) values are given.

I am certainly grateful to the authors for their effort and the communication of the data. Their work has a lot of merit, but I think it should be reformulated as a case report.

Author Response

Comments review 1:

I am grateful to have had the opportunity to read carefully this interesting object of study. It is very stimulating to propose the practice of exercise to any type of patient, no matter how affected they may be. It is interesting as communications of several cases, but I think the conclusions should either be removed or changed, because there are many methodological problems to reach any conclusion, and put hypothetical conclusions make the work loses strength. I would communicate only the data for the readers' assessment.

Authors: Thank you, we have revised the paper according to your inputs.

First of all, I would remove "Exercise Training" from the title, and leave only: "home-based physical exercise" because the essential characteristics of a training program such as individualization and intensity of the load are not communicated. There is hardly any mention of frequency and time of exercise (with many limitations), nor is the type and mode of exercise specified with sufficient rigor.

Authors: Thank you for your comment. We have included further details on the intervention. We have used this type of intervention in a broad of studies in different clinical populations [19,20]. The exercise intensity was assessed through the Children’s OMNI scale of perceived subject exertion (PSE) [21]. Progression was carried out every 4 weeks using the OMNI scale (week 1-4, PSE: 5-6; week 4-8, PSE: 6-7 and week 8-12, PSE: 7-8), modifying the number of repetitions (10 to 15), sets (3 to 4), and/or duration of the sets (30 to 45 seconds). The progression occurred prudently in accordance with the patient’s tolerance to the exercise program, since we were dealing with a rare and new condition, so that we did not know any possible clinical manifestations or adverse events that could emerge from the intervention. Patients received instructional videos, photos, and “gifs” describing the exercise program. A video call was conducted prior to program initiation to provide details on the program and collect information on patient’s health status.

References:

[19]      Astley C, Clemente G, Terreri MT, Carneiro CG, Lima MS, Buchpiguel CA, et al. Home-Based Exercise Training in Childhood-Onset Takayasu Arteritis: A Multicenter, Randomized, Controlled Trial. Front Immunol 2021;12:1–10. https://doi.org/10.3389/fimmu.2021.705250.

[20]     Sieczkowska SM, Astley C, Marques IG, Iraha AY, Franco TC, Ihara BP, et al. A home-based exercise program during COVID-19 pandemic: Perceptions and acceptability of juvenile systemic lupus erythematosus and juvenile idiopathic arthritis adolescents. Lupus 2022;31. https://doi.org/10.1177/09612033221083273.

[21]     Robertson RJ, Goss FL, Boer NF, Peoples JA, Foreman AJ, Dabayebeh IM, et al. Children’s OMNI scale of perceived exertion: Mixed gender and race validation. Med Sci Sports Exerc 2000;32. https://doi.org/10.1097/00005768-200002000-00029.

The reasons for leaving only the communication of data without interpreting them in the form of conclusions is because there were very few patients, it is not known why some did not want to follow the program (were they more affected?), it is not clear but it seems that there are different lags for each patient from the time they become ill until the tests are performed. It is not explained whether or not the patients who died were doing home exercise. The clinical significance of the observed changes is not explained, nor is the interference of medications. Nor is there any statistical treatment, despite the fact that p (?) values are given. I am certainly grateful to the authors for their effort and the communication of the data. Their work has a lot of merit, but I think it should be reformulated as a case report.

Authors: We have provided more details on the flux of participants. Three patients did not participate in the study because, as reported by their parents, they had difficulties to adhere to online exercise sessions. The reviewer in noting different lags for each patient being included in the study. As MIS-C is a rare condition related to SARS-CoV-2 infection in children, the flux of new participants was continuous. The patients who died was hospitalized, and this was before the exercise training intervention. We compared the frequency of abnormal < 2, borderline between 2 and 2.5 and normal when > 2.5 coronary flow reserve using Fisher’s exact test. All these issues have been addressed in the new version of the manuscript. Thank you for your comments and inputs.

Reviewer 2 Report

In the manuscript, “Exercise Training in the Recovery of Multisystem Inflammatory Syndrome in Children: A Case Series Study”- Astley et al, suggested that exercise plays a great role in the treatment of post-discharge MIS-C patients. The study is well performed, the flow of writing is excellent. Please follow the comments to further improve the manuscript.

In the materials and method section, lines 82-83, Please mention the catalog number and company name for the ELISA kit.

Author Response

Comments review 2:

In the manuscript, “Exercise Training in the Recovery of Multisystem Inflammatory Syndrome in Children: A Case Series Study”- Astley et al, suggested that exercise plays a great role in the treatment of post-discharge MIS-C patients. The study is well performed, the flow of writing is excellent. Please follow the comments to further improve the manuscript.

Authors: Thank you, we have revised the paper according to your inputs.

In the materials and method section, lines 82-83, Please mention the catalog number and company name for the ELISA kit.

Authors: We have included this information in the new version of the manuscript.

Reviewer 3 Report

This study aimed to assess a 12-wk, home-based exercise intervention in children and adolescents after MIS-C diagnosis. The manuscript required improvements.

Introduction: Incomplete. The authors nothing stated about the effects of different types of exercise in MIS-C complications and the relations between exercise response and blood biochemical markers. What’s the mechanisms behind multisystem inflammatory syndrome in children that can be prevented by exercise? What type of exercise (aerobic, anaerobic, resistance…)? Additionally, the link between Covid-19 Exercise and multisystem inflammatory syndrome in children is still missing. The introduction should be improved with more information.

Methods:
Study design and patients -> Include a flowchart for participants.
Table 1 only appear in the results section. I suggest to breakdown table 1 in two. Methods related information and results related information.
Statistics: No comparisons between baseline and after 12w? Nothing presented in methods section regarding the statistical analysis (and p values are presented in results).
Exercise program: Nothing is presented about the exercise prescription based on intensity and effort. Which lead me to assume that it was not controlled. So, this is a physical activity program?

Results:
Most of the more important results are in supplementary material which should be in the paper. This study is based on specific cases. So, all the information is important.
How did the authors get the radar plots? It is not presented in methods. The paper is incomplete.

Discussion:

Lacks comparison with improvements in similar studies (% of change) per example.
It seems that the authors misunderstand the concepts of exercise and physical activity.
Limitation and strengths are well constructed.

Conclusion:
Adequate.

Author Response

Comments review 3:

This study aimed to assess a 12-wk, home-based exercise intervention in children and adolescents after MIS-C diagnosis. The manuscript required improvements.

Introduction: Incomplete. The authors nothing stated about the effects of different types of exercise in MIS-C complications and the relations between exercise response and blood biochemical markers. What’s the mechanisms behind multisystem inflammatory syndrome in children that can be prevented by exercise? What type of exercise (aerobic, anaerobic, resistance…)? Additionally, the link between Covid-19 Exercise and multisystem inflammatory syndrome in children is still missing. The introduction should be improved with more information.

Authors: Thank you, we have revised the paper according to your inputs.

Methods: Study design and patients -> Include a flowchart for participants.
Table 1 only appear in the results section. I suggest to breakdown table 1 in two.

Authors: We have included the flowchart accordingly.

Methods related information and results related information. 
Statistics: No comparisons between baseline and after 12w? Nothing presented in methods section regarding the statistical analysis (and p values are presented in results).
Authors: Pre-to-post statistics was not conducted since this is deemed inappropriate in case studies as ours¹. However, we did compare the frequency of abnormal values (< 2, borderline between 2 and 2.5 and normal when > 2.5) for coronary flow reserve using Fisher’s exact test. We have added this information in the new version of the manuscript.

¹ Carey TS, Boden SD. A critical guide to case series reports. Spine. 2003 Aug 1;28(15):1631-4. doi: 10.1097/01.BRS.0000083174.84050.E5.

Exercise program: Nothing is presented about the exercise prescription based on intensity and effort. Which lead me to assume that it was not controlled. So, this is a physical activity program?

Authors: Thank you for your comment. We have included further details on the intervention. We have used this type of intervention in a broad of studies in different clinical populations [19,20].  The exercise intensity was assessed through the Children’s OMNI scale of perceived subject exertion (PSE) [21]. Progression was carried out every 4 weeks using the OMNI scale (week 1-4, PSE: 5-6; week 4-8, PSE: 6-7 and week 8-12, PSE: 7-8), modifying the number of repetitions (10 to 15), sets (3 to 4), and/or duration of the sets (30 to 45 seconds). The progression occurred prudently in accordance with the patient’s tolerance to the exercise program, since we were dealing with a rare and new condition, so that we did not know any possible clinical manifestations or adverse events that could emerge from the intervention. Patients received instructional videos, photos, and “gifs” describing the exercise program. A video call was conducted prior to program initiation to provide details on the program and collect information on patient’s health status.

References:

[19]      Astley C, Clemente G, Terreri MT, Carneiro CG, Lima MS, Buchpiguel CA, et al. Home-Based Exercise Training in Childhood-Onset Takayasu Arteritis: A Multicenter, Randomized, Controlled Trial. Front Immunol 2021;12:1–10. https://doi.org/10.3389/fimmu.2021.705250.

[20]     Sieczkowska SM, Astley C, Marques IG, Iraha AY, Franco TC, Ihara BP, et al. A home-based exercise program during COVID-19 pandemic: Perceptions and acceptability of juvenile systemic lupus erythematosus and juvenile idiopathic arthritis adolescents. Lupus 2022;31. https://doi.org/10.1177/09612033221083273.

[21]     Robertson RJ, Goss FL, Boer NF, Peoples JA, Foreman AJ, Dabayebeh IM, et al. Children’s OMNI scale of perceived exertion: Mixed gender and race validation. Med Sci Sports Exerc 2000;32. https://doi.org/10.1097/00005768-200002000-00029.

Results: Most of the more important results are in supplementary material which should be in the paper. This study is based on specific cases. So, all the information is important.
How did the authors get the radar plots? It is not presented in methods. The paper is incomplete.

Authors: Thank you for your comment. We have added data initially displayed in supplementary material within the main manuscript. Radar plot was created using absolute values from pre and post periods for each patient and each PODCI domains (upper extremity and physical functioning, transfer and basic mobility, sports and physical function, pain/comfort and happiness). All radar plots were generate using fmsb package (v. 0.7.5; Nakazawa, M., 2023) in the environment R (version 3.5.3; R Core Team 2020). This information has been included in the new version of the manuscript (Pediatric Outcomes Data Collection Instrument PODCI section).

Discussion: Lacks comparison with improvements in similar studies (% of change) per example.
It seems that the authors misunderstand the concepts of exercise and physical activity.
Limitation and strengths are well constructed.

Authors: We appreciate your suggestion, but contrasting the data to the literature was difficult since this is, to our knowledge, the first study involving exercise intervention in MIS-c patients.

Conclusion:
Adequate.

Authors: Thank you for your comments.

Round 2

Reviewer 3 Report

.